# Bayesian Semi-supervised Learning with Graph Gaussian Processes

**Yin Cheng Ng**[1]**, Nicolò Colombo**[1]**, Ricardo Silva**[1,2]
[1]Statistical Science, University College London
[2]The Alan Turing Institute
{y.ng.12, nicolo.colombo, ricardo.silva}@ucl.ac.uk

## Abstract

We propose a data-efficient Gaussian process-based Bayesian approach to the semi-supervised learning problem on graphs. The proposed model shows extremely competitive performance when compared to the state-of-the-art graph neural networks on semi-supervised learning benchmark experiments, and outperforms the neural networks in active learning experiments where labels are scarce. Furthermore, the model does not require a validation data set for early stopping to control over-fitting. Our model can be viewed as an instance of empirical distribution regression weighted locally by network connectivity. We further motivate the intuitive construction of the model with a Bayesian linear model interpretation where the node features are filtered by an operator related to the graph Laplacian. The method can be easily implemented by adapting off-the-shelf scalable variational inference algorithms for Gaussian processes.

## 1 Introduction

Data sets with network and graph structures that describe the relationships between the data points (nodes) are abundant in the real world. Examples of such data sets include friendship graphs on social networks, citation networks of academic papers, web graphs and many others. The relational graphs often provide rich information in addition to the node features that can be exploited to build better predictive models of the node labels, which can be costly to collect. In scenarios where there are not enough resources to collect sufficient labels, it is important to design data-efficient models that can generalize well with few training labels. The class of learning problems where a relational graph of the data points is available is referred to as graph-based semi-supervised learning in the literature [7, 47].

Many of the successful graph-based semi-supervised learning models are based on graph Laplacian regularization or learning embeddings of the nodes. While these models have been widely adopted, their predictive performance leaves room for improvement. More recently, powerful graph neural networks that surpass Laplacian and embedding based methods in predictive performance have become popular. However, neural network models require relatively larger number of labels to prevent over-fitting and work well. We discuss the existing models for graph-based semi-supervised learning in detail in Section 4.

We propose a new Gaussian process model for graph-based semi-supervised learning problems that can generalize well with few labels, bridging the gap between the simpler models and the more data intensive graph neural networks. The proposed model is also competitive with graph neural networks in settings where there are sufficient labelled data. While posterior inference for the proposed model is intractable for classification problems, scalable variational inducing point approximation method for Gaussian processes can be directly applied to perform inference. Despite the potentially large number of inducing points that need to be optimized, the model is protected from over-fitting by the

variational lower bound, and does not require a validation data set for early stopping. We refer to the proposed model as the graph Gaussian process (GGP).

## 2 Background

In this section, we briefly review key concepts in Gaussian processes and the relevant variational approximation technique. Additionally, we review the graph Laplacian, which is relevant to the alternative view of the model that we describe in Section 3.1. This section also introduces the notation used across the paper.

### 2.1 Gaussian Processes

A Gaussian process $f(\mathbf{x})$ (GP) is an infinite collection of random variables, of which any finite subset is jointly Gaussian distributed. Consequently, a GP is completely specified by its mean function $m(\mathbf{x})$ and covariance kernel function $k_\theta(\mathbf{x}, \mathbf{x}')$, where $\mathbf{x}, \mathbf{x}' \in \mathcal{X}$ denote the possible inputs that index the GP and $\theta$ is a set of hyper-parameters parameterizing the kernel function. We denote the GP as follows

$$f(\mathbf{x}) \sim \mathcal{GP}\big(m(\mathbf{x}), k_\theta(\mathbf{x}, \mathbf{x}')\big). \tag{1}$$

GPs are widely used as priors on functions in the Bayesian machine learning literatures because of their wide support, posterior consistency, tractable posterior in certain settings and many other good properties. Combined with a suitable likelihood function as specified in Equation 2, one can construct a regression or classification model that probabilistically accounts for uncertainties and control over-fitting through Bayesian smoothing. However, if the likelihood is non-Gaussian, such as in the case of classification, inferring the posterior process is analytically intractable and requires approximations. The GP is connected to the observed data via the likelihood function

$$y_n \mid f(\mathbf{x}_n) \sim p(y_n|f(\mathbf{x}_n)) \quad \forall n \in \{1, \dots, N\}. \tag{2}$$

The positive definite kernel function $k_\theta(\mathbf{x}, \mathbf{x}') : \mathcal{X} \times \mathcal{X} \to \mathbb{R}$ is a key component of GP that specifies the covariance of $f(\mathbf{x})$ *a priori*. While $k_\theta(\mathbf{x}, \mathbf{x}')$ is typically directly specified, any kernel function can be expressed as the inner product of features maps $\langle \phi(\mathbf{x}), \phi(\mathbf{x}') \rangle_{\mathcal{H}}$ in the Hilbert space $\mathcal{H}$. The dependency of the feature map on $\theta$ is implicitly assumed for conciseness. The feature map $\phi(\mathbf{x}) : \mathcal{X} \to \mathcal{H}$ projects $\mathbf{x}$ into a typically high-dimensional (possibly infinite) feature space such that linear models in the feature space can model the target variable $y$ effectively. Therefore, GP can equivalently be formulated as

$$f(\mathbf{x}) = \phi(\mathbf{x})^\mathsf{T}\mathbf{w}, \tag{3}$$

where $\mathbf{w}$ is assigned a multivariate Gaussian prior distribution and marginalized. In this paper, we assume the index set to be $\mathcal{X} = \mathbb{R}^{D \times 1}$ without loss of generality.

For a detailed review of the GP and the kernel functions, please refer to [45].

### 2.1.1 Scalable Variational Inference for GP

Despite the flexibility of the GP prior, there are two major drawbacks that plague the model. First, if the likelihood function in Equation 2 is non-Gaussian, posterior inference cannot be computed analytically. Secondly, the computational complexity of the inference algorithm is $O(N^3)$ where $N$ is the number of training data points, rendering the model inapplicable to large data sets.

Fortunately, modern variational inference provides a solution to both problems by introducing a set of $M$ inducing points $\mathbf{Z} = [\mathbf{z_1}, \dots, \mathbf{z}_M]^\mathsf{T}$, where $\mathbf{z}_m \in \mathbb{R}^{D \times 1}$. The inducing points, which are variational parameters, index a set of random variables $\mathbf{u} = [f(\mathbf{z_1}), \dots, f(\mathbf{z}_M)]^\mathsf{T}$ that is a subset of the GP function $f(\mathbf{x})$. Through conditioning and assuming $m(\mathbf{x})$ is zero, the conditional GP can be expressed as

$$f(\mathbf{x}) \mid \mathbf{u} \sim \mathcal{GP}(\mathbf{k}_{\mathbf{zx}}^\mathsf{T}\mathbf{K}_{\mathbf{zz}}^{-1}\mathbf{u}, k_\theta(\mathbf{x}, \mathbf{x}) - \mathbf{k}_{\mathbf{zx}}^\mathsf{T}\mathbf{K}_{\mathbf{zz}}^{-1}\mathbf{k}_{\mathbf{zx}}) \tag{4}$$

where $\mathbf{k}_{\mathbf{zx}} = [k_\theta(\mathbf{z_1}, \mathbf{x}), \dots, k_\theta(\mathbf{z}_M, \mathbf{x})]$ and $[\mathbf{K}_{\mathbf{zz}}]_{ij} = k_\theta(\mathbf{z}_i, \mathbf{z}_j)$. Naturally, $p(\mathbf{u}) = \mathcal{N}(\mathbf{0}, \mathbf{K}_{\mathbf{zz}})$. The variational posterior distribution of $\mathbf{u}$, $q(\mathbf{u})$ is assumed to be a multivariate Gaussian distribution with mean $\mathbf{m}$ and covariance matrix $\mathbf{S}$. Following the standard derivation of variational inference, the Evidence Lower Bound (ELBO) objective function is

$$\mathcal{L}(\theta, \mathbf{Z}, \mathbf{m}, \mathbf{S}) = \sum_{n=1}^{N} \mathbb{E}_{q(f(\mathbf{x}_n))}[\log p(y_n|f(\mathbf{x}_n))] - \mathrm{KL}[q(\mathbf{u})||p(\mathbf{u})]. \tag{5}$$

The variational distribution $q(f(\mathbf{x}_n))$ can be easily derived from the conditional GP in Equation 4 and $q(\mathbf{u})$, and its expectation can be approximated effectively using 1-dimensional quadratures. We refer the readers to [30] for detailed derivations and results.

## 2.2 The Graph Laplacian

Given adjacency matrix $\mathbf{A} \in \{0,1\}^{N \times N}$ of an undirected binary graph $\mathcal{G} = (\mathcal{V}, \mathcal{E})$ without self-loop, the corresponding graph Laplacian is defined as

$$\mathbf{L} = \mathbf{D} - \mathbf{A}, \tag{6}$$

where $\mathbf{D}$ is the $N \times N$ diagonal node degree matrix. The graph Laplacian can be viewed as an operator on the space of functions $g : \mathcal{V} \to \mathbb{R}$ indexed by the graph's nodes such that

$$\mathbf{L}g(n) = \sum_{v \in Ne(n)} [g(n) - g(v)], \tag{7}$$

where $Ne(n)$ is the set containing neighbours of node $n$. Intuitively, applying the Laplacian operator to the function $g$ results in a function that quantifies the variability of $g$ around the nodes in the graph.

The Laplacian's spectrum encodes the geometric properties of the graph that are useful in crafting graph filters and kernels [37, 43, 4, 9]. As the Laplacian matrix is real symmetric and diagonalizable, its eigen-decomposition exists. We denote the decomposition as

$$\mathbf{L} = \mathbf{U}\mathbf{\Lambda}\mathbf{U}^\mathsf{T}, \tag{8}$$

where the columns of $\mathbf{U} \in \mathbb{R}^{N \times N}$ are the eigenfunctions of $\mathbf{L}$ and the diagonal $\mathbf{\Lambda} \in \mathbb{R}^{N \times N}$ contains the corresponding eigenvalues. Therefore, the Laplacian operator can also be viewed as a filter on function $g$ re-expressed using the eigenfunction basis. Regularization can be achieved by directly manipulating the eigenvalues of the system [39]. We refer the readers to [4, 37, 9] for comprehensive reviews of the graph Laplacian and its spectrum.

# 3 Graph Gaussian Processes

Given a data set of size $N$ with $D$-dimensional features $\mathbf{X} = [\mathbf{x}_1, \ldots, \mathbf{x}_N]^\mathsf{T}$, a symmetric binary adjacency matrix $\mathbf{A} \in \{0,1\}^{N \times N}$ that represents the relational graph of the data points and labels for a subset of the data points, $\mathcal{Y}_o = [y_1, \ldots, y_O]$, with each $y_i \in \{1, \ldots, K\}$, we seek to predict the unobserved labels of the remaining data points $\mathcal{Y}_U = [y_{O+1}, \ldots y_N]$. We denote the set of all labels as $\mathbf{Y} = \mathcal{Y}_O \cup \mathcal{Y}_U$.

The GGP specifies the conditional distribution $p_\theta(\mathbf{Y}|\mathbf{X}, \mathbf{A})$, and predicts $\mathcal{Y}_U$ via the predictive distribution $p_\theta(\mathcal{Y}_U|\mathcal{Y}_O, \mathbf{X}, \mathbf{A})$. The joint model is specified as the product of the conditionally independent likelihood $p(y_n|h_n)$ and the GGP prior $p_\theta(\mathbf{h}|\mathbf{X}, \mathbf{A})$ with hyper-parameters $\theta$. The latent likelihood parameter vector $\mathbf{h} \in \mathbb{R}^{N \times 1}$ is defined in the next paragraph.

First, the model factorizes as

$$p_\theta(\mathbf{Y}, \mathbf{h}|\mathbf{X}, \mathbf{A}) = p_\theta(\mathbf{h}|\mathbf{X}, \mathbf{A}) \prod_{n=1}^{N} p(y_n|h_n), \tag{9}$$

where for the multi-class classification problem that we are interested in, $p(y_n \mid h_n)$ is given by the robust-max likelihood [30, 16, 23, 21, 20].

Next, we construct the GGP prior from a Gaussian process distributed latent function $f(\mathbf{x}) : \mathbb{R}^{D \times 1} \to \mathbb{R}$, $f(\mathbf{x}) \sim \mathcal{GP}(0, k_\theta(\mathbf{x}, \mathbf{x}'))$, where the *key assumption* is that the likelihood parameter $h_n$ for data point $n$ is an average of the values of $f$ over its 1-hop neighbourhood $Ne(n)$ as given by $\mathbf{A}$:

$$h_n = \frac{f(\mathbf{x}_n) + \sum_{l \in Ne(n)} f(\mathbf{x}_l)}{1 + D_n} \tag{10}$$

where $Ne(n) = \{l : l \in \{1, \ldots, N\}, \mathbf{A}_{nl} = 1\}$, $D_n = |Ne(n)|$. We further motivate this key assumption in Section 3.1.

As $f(\mathbf{x})$ has a zero mean function, the GGP prior can be succinctly expressed as a multivariate Gaussian random field

$$p_\theta(\mathbf{h}|\mathbf{X}, \mathbf{A}) = \mathcal{N}(\mathbf{0}, \mathbf{P}\mathbf{K_{XX}}\mathbf{P}^\mathsf{T}), \qquad (11)$$

where $\mathbf{P} = (\mathbf{I} + \mathbf{D})^{-1}(\mathbf{I} + \mathbf{A})$ and $[\mathbf{K_{XX}}]_{ij} = k_\theta(\mathbf{x}_i, \mathbf{x}_j)$. A suitable kernel function $k_\theta(\mathbf{x}_i, \mathbf{x}_j)$ for the task at hand can be chosen from the suite of well-studied existing kernels, such as those described in [13]. We refer to the chosen kernel function as the *base kernel* of the GGP. The $\mathbf{P}$ matrix is sometimes known as the random-walk matrix in the literatures [9]. A graphical model representation of the proposed model is shown in Figure 1.

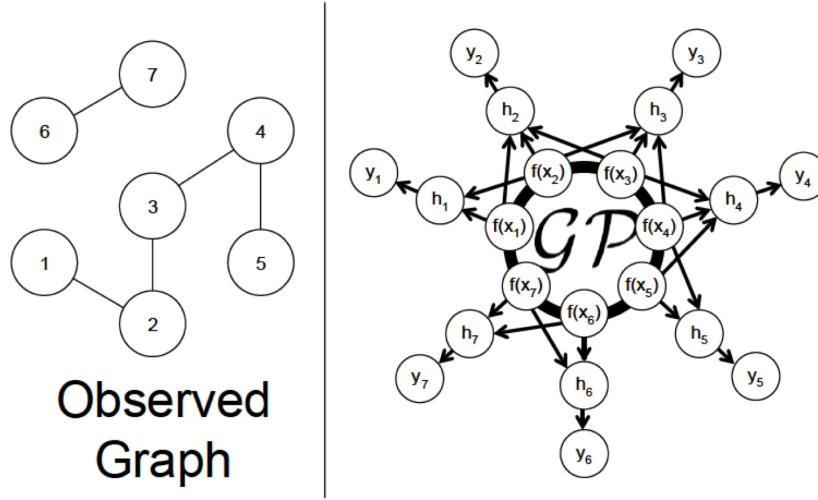

Figure 1: The figure depicts a relational graph (left) and the corresponding GGP represented as a graphical model (right). The thick circle represents a set of fully connected nodes.

The covariance structure specified in Equation 11 is equivalent to the pairwise covariance

$$Cov(h_m, h_n) = \frac{1}{(1+D_m)(1+D_n)} \sum_{i \in \{m \cup Ne(m)\}} \sum_{j \in \{n \cup Ne(n)\}} k_\theta(\mathbf{x}_i, \mathbf{x}_j)$$

$$= \langle \frac{1}{1+D_m} \sum_{i \in \{m \cup Ne(m)\}} \phi(\mathbf{x}_i), \frac{1}{1+D_n} \sum_{j \in \{n \cup Ne(n)\}} \phi(\mathbf{x}_j) \rangle_{\mathcal{H}} \qquad (12)$$

where $\phi(\cdot)$ is the feature map that corresponds to the base kernel $k_\theta(\cdot, \cdot)$. Equation 12 can be viewed as the inner product between the empirical kernel mean embeddings that correspond to the bags of node features observed in the 1-hop neighborhood sub-graphs of node $m$ and $n$, relating the proposed model to the Gaussian process distribution regression model presented in e.g. [15].

More specifically, we can view the GGP as a distribution classification model for the labelled bags of node features $\{(\{\mathbf{x}_i | i \in \{n \cup Ne(n)\}\}, y_n)\}_{n=1}^{O}$, such that the unobserved distribution $P_n$ that generates $\{\mathbf{x}_i | i \in \{n \cup Ne(n)\}\}$ is summarized by its empirical kernel mean embedding

$$\hat{\mu}_n = \frac{1}{1+D_n} \sum_{j \in \{n \cup Ne(n)\}} \phi(\mathbf{x}_j). \qquad (13)$$

The prior on $\mathbf{h}$ can equivalently be expressed as $\mathbf{h} \sim \mathcal{GP}(0, \langle \hat{\mu}_m, \hat{\mu}_n \rangle_{\mathcal{H}})$. For detailed reviews of the kernel mean embedding and distribution regression models, we refer the readers to [32] and [41] respectively.

One main assumption of the 1-hop neighbourhood averaging mechanism is homophily - i.e., nodes with similar covariates are more likely to form connections with each others [17]. The assumption allows us to approximately treat the node covariates from a 1-hop neighbourhood as samples drawn from the same data distribution, in order to model them using distribution regression. While it is perfectly reasonable to consider multi-hops neighbourhood averaging, the homophily assumption

starts to break down if we consider 2-hop neighbours which are not directly connected. Nevertheless, it is interesting to explore non-naive ways to account for multi-hop neighbours in the future, such as stacking 1-hop averaging graph GPs in a structure similar to that of the deep Gaussian processes [10, 34], or having multiple latent GPs for neighbours of different hops that are summed up in the likelihood functions.

## 3.1 An Alternative View of GGP

In this section, we present an alternative formulation of the GGP, which results in an intuitive interpretation of the model. The alternative formulation views the GGP as a Bayesian linear model on feature maps of the nodes that have been transformed by a function related to the graph Laplacian $\mathbf{L}$.

As we reviewed in Section 2.1, the kernel matrix $\mathbf{K_{XX}}$ in Equation 11 can be written as the product of feature map matrix $\Phi_\mathbf{X}\Phi_\mathbf{X}^\mathsf{T}$ where row $n$ of $\Phi_\mathbf{X}$ corresponds to the feature maps of node $n$, $\phi(\mathbf{x}_n) = [\phi_{n1}, \ldots, \phi_{nQ}]^\mathsf{T}$. Therefore, the covariance matrix in Equation 11, $\mathbf{P}\Phi_\mathbf{X}\Phi_\mathbf{X}^\mathsf{T}\mathbf{P}^\mathsf{T}$, can be viewed as the product of the transformed feature maps

$$\widehat{\Phi}_\mathbf{X} = \mathbf{P}\Phi_\mathbf{X} = (\mathbf{I} + \mathbf{D})^{-1}\mathbf{D}\Phi_\mathbf{X} + (\mathbf{I} + \mathbf{D})^{-1}(\mathbf{I} - \mathbf{L})\Phi_\mathbf{X}. \tag{14}$$

where $\mathbf{L}$ is the graph Laplacian matrix as defined in Equation 6. Isolating the transformed feature maps for node $n$ (i.e., row $n$ of $\widehat{\Phi}_\mathbf{X}$) gives

$$\hat{\phi}(\mathbf{x}_n) = \frac{D_n}{1 + D_n}\phi(\mathbf{x}_n) + \frac{1}{1 + D_n}[(\mathbf{I} - \mathbf{L})\Phi_\mathbf{X}]_n^\mathsf{T}, \tag{15}$$

where $D_n$ is the degree of node $n$ and $[\cdot]_n$ denotes row $n$ of a matrix. The proposed GGP model is equivalent to a supervised Bayesian linear classification model with a feature pre-processing step that follows from the expression in Equation 15. For isolated nodes ($D_n = 0$), the expression in Equation 15 leaves the node feature maps unchanged ($\hat{\phi} = \phi$).

The $(\mathbf{I} - \mathbf{L})$ term in Equation 15 can be viewed as a spectral filter $\mathbf{U}(\mathbf{I} - \Lambda)\mathbf{U}^\mathsf{T}$, where $\mathbf{U}$ and $\Lambda$ are the eigenmatrix and eigenvalues of the Laplacian as defined in Section 2.2. For connected nodes, the expression results in new features that are weighted averages of the original features and features transformed by the spectral filter. The alternative formulation opens up opportunities to design other spectral filters with different regularization properties, such as those described in [39], that can replace the $(\mathbf{I} - \mathbf{L})$ expression in Equation 15. We leave the exploration of this research direction to future work.

In addition, it is well-known that many graphs and networks observed in the real world follow the power-law node degree distributions [17], implying that there are a handful of nodes with very large degrees (known as hubs) and many with relatively small numbers of connections. The nodes with few connections (small $D_n$) are likely to be connected to one of the handful of heavily connected nodes, and their transformed node feature maps are highly influenced by the features of the hub nodes. On the other hand, individual neighbours of the hub nodes have relatively small impact on the hub nodes because of the large number of neighbours that the hubs are connected to. This highlights the asymmetric outsize influence of hubs in the proposed GGP model, such that a mis-labelled hub node may result in a more significant drop in the model's accuracy compared to a mis-labelled node with much lower degree of connections.

## 3.2 Variational Inference with Inducing Points

Posterior inference for the GGP is analytically intractable because of the non-conjugate likelihood. We approximate the posterior of the GGP using a variational inference algorithm with inducing points similar to the inter-domain inference algorithm presented in [42]. Implementing the GGP with its variational inference algorithm amounts to implementing a new kernel function that follows Equation 12 in the GPflow Python package.[1]

We introduce a set of $M$ inducing random variables $\mathbf{u} = [f(\mathbf{z}_1), \ldots, f(\mathbf{z}_M)]^\mathsf{T}$ indexed by inducing points $\{\mathbf{z}_m\}_{m=1}^M$ in the same domain as the GP function $f(\mathbf{x}) \sim \mathcal{GP}(\mathbf{0}, k_\theta(\mathbf{x}, \mathbf{x}'))$. As a result, the

inter-domain covariance between $h_n$ and $f(\mathbf{z}_m)$ is

$$Cov(h_n, f(\mathbf{z}_m)) = \frac{1}{D_n + 1}\Big[k_\theta(\mathbf{x}_n, \mathbf{z}_m) + \sum_{l \in Ne(n)} k_\theta(\mathbf{x}_l, \mathbf{z}_m)\Big]. \qquad (16)$$

Additionally, we introduce a multivariate Gaussian variational distribution $q(\mathbf{u}) = \mathcal{N}(\mathbf{m}, \mathbf{SS}^\mathsf{T})$ for the inducing random variables with variational parameters $\mathbf{m} \in \mathbb{R}^{M \times 1}$ and the lower triangular $\mathbf{S} \in \mathbb{R}^{M \times M}$. Through Gaussian conditioning, $q(\mathbf{u})$ results in the variational Gaussian distribution $q(\mathbf{h})$ that is of our interest. The variational parameters $\mathbf{m}, \mathbf{S}, \{\mathbf{z}_m\}_{m=1}^M$ and the kernel hyper-parameters $\theta$ are then jointly fitted by maximizing the ELBO function in Equation 5.

### 3.3 Computational Complexity

The computational complexity of the inference algorithm is $O(|\mathcal{Y}_o|M^2)$. In the experiments, we chose $M$ to be the number of labelled nodes in the graph $|\mathcal{Y}_o|$, which is small relative to the total number of nodes. Computing the covariance function in Equation 12 incurs a computational cost of $O(D_{max}^2)$ per labelled node, where $D_{max}$ is the maximum node degree. In practice, the computational cost of computing the covariance function is small because of the sparse property of graphs typically observed in the real-world [17].

## 4 Related Work

Graph-based learning problems have been studied extensively by researchers from both machine learning and signal processing communities, leading to many models and algorithms that are well-summarized in review papers [4, 35, 37].

Gaussian process-based models that operate on graphs have previously been developed in the closely related relational learning discipline, resulting in the mixed graph Gaussian process (XGP) [38] and relational Gaussian process (RGP) [8]. Additionally, the renowned Label Propagation (LP)[48] model can also be viewed as a GP with its covariance structure specified by the graph Laplacian matrix [49]. The GGP differs from the previously proposed GP models in that the local neighbourhood structures of the graph and the node features are directly used in the specification of the covariance function, resulting in a simple model that is highly effective.

Models based on Laplacian regularization that restrict the node labels to vary smoothly over graphs have also been proposed previously. The LP model can be viewed as an instance under this framework. Other Laplacian regularization based models include the deep semi-supervised embedding [44] and the manifold regularization [3] models. As shown in the experimental results in Table 1, the predictive performance of these models fall short of other more sophisticated models.

Additionally, models that extract embeddings of nodes and local sub-graphs which can be used for predictions have also been proposed by multiple authors. These models include DeepWalk [33], node2vec [19], planetoid [46] and many others. The proposed GGP is related to the embedding based models in that it can be viewed as a GP classifer that takes empirical kernel mean embeddings extracted from the 1-hop neighbourhood sub-graphs as inputs to predict node labels.

Finally, many geometric deep learning models that operate on graphs have been proposed and shown to be successful in graph-based semi-supervised learning problems. The earlier models including [26, 36, 18] are inspired by the recurrent neural networks. On the other hand, convolution neural networks that learn convolutional filters in the graph Laplacian spectral domain have been demonstrated to perform well. These models include the spectral CNN [5], DCNN [1], ChebNet [12] and GCN [25]. Neural networks that operate on the graph spectral domain are limited by the graph-specific Fourier basis. The more recently proposed MoNet [31] addressed the graph-specific limitation of spectral graph neural networks. The idea of filtering in graph spectral domain is a powerful one that has also been explored in the kernel literatures [39, 43]. We draw parallels between our proposed model and the spectral filtering approaches in Section 3.1, where we view the GGP as a standard GP classifier operating on feature maps that have been transformed through a filter that can be related to the graph spectral domain.

Our work has also been inspired by literatures in Gaussian processes that mix GPs via an additive function, such as [6, 14, 42].

# 5  Experiments

We present two sets of experiments to benchmark the predictive performance of the GGP against existing models under two different settings. In Section 5.1, we demonstrate that the GGP is a viable and extremely competitive alternative to the graph convolutional neural network (GCN) in settings where there are sufficient labelled data points. In Section 5.2, we test the models in an active learning experimental setup, and show that the GGP outperforms the baseline models when there are few training labels.

## 5.1  Semi-supervised Classification on Graphs

The semi-supervised classification experiments in this section exactly replicate the experimental setup in [25], where the GCN is known to perform well. The three benchmark data sets, as described in Table 2, are citation networks with bag-of-words (BOW) features, and the prediction targets are the topics of the scientific papers in the citation networks.

The experimental results are presented in Table 1, and show that the predictive performance of the proposed GGP is competitive with the GCN and MoNet [31] (another deep learning model), and superior to the other baseline models. While the GCN outperforms the proposed model by small margins on the test sets with $1,000$ data points, it is important to note that the GCN had access to 500 additional labelled data points for early stopping. As the GGP does not require early stopping, the additional labelled data points can instead be directly used to train the model to significantly improve the predictive performance. To demonstrate this advantage, we report another set of results for a GGP trained using the 500 additional data points in Table 1, in the row labelled as '**GGP-X**'. The boost in the predictive performances shows that the GGP can better exploit the available labelled data to make predictions.

The GGP base kernel of choice is the 3rd degree polynomial kernel, which is known to work well with high-dimensional BOW features [45]. We re-weighed the BOW features using the popular term frequency-inverse document frequency (TFIDF) technique [40]. The variational parameters and the hyper-parameters were jointly optimized using the ADAM optimizer [24]. The baseline models that we compared to are the ones that were also presented and compared to in [25] and [31].

|              | Cora  | Citeseer | Pubmed |
|--------------|-------|----------|--------|
| GGP          | 80.9% | 69.7%    | 77.1%  |
| GGP-X        | 84.7% | 75.6%    | 82.4%  |
| GCN[25]      | 81.5% | 70.3%    | 79.0%  |
| DCNN[1]      | 76.8% | -        | 73.0%  |
| MoNet[31]    | 81.7% | -        | 78.8%  |
| DeepWalk[33] | 67.2% | 43.2%    | 65.3%  |
| Planetoid[46]| 75.7% | 64.7%    | 77.2%  |
| ICA[27]      | 75.1% | 69.1%    | 73.9%  |
| LP[48]       | 68.0% | 45.3%    | 63.0%  |
| SemiEmb[44]  | 59.0% | 59.6%    | 71.1%  |
| ManiReg[3]   | 59.5% | 60.1%    | 70.7%  |

Table 1: This table shows the test classification accuracies of the semi-supervised learning experiments described in Section 5.1. The test sets consist of $1,000$ data points. The GGP accuracies are averaged over 10 random restarts. The results for DCNN and MoNet are copied from [31] while the results for the other models are from [25]. Please refer to Section 5.1 for discussions of the results.

|          | Type     | $N_{nodes}$ | $N_{edges}$ | $N_{label\_cat.}$ | $D_{features}$ | Label Rate |
|----------|----------|-------------|-------------|-------------------|----------------|------------|
| **Cora**     | Citation | $2,708$     | $5,429$     | 7                 | $1,433$        | 0.052      |
| **Citeseer** | Citation | $3,327$     | $4,732$     | 6                 | $3,703$        | 0.036      |
| **Pubmed**   | Citation | $19,717$    | $44,338$    | 3                 | $500$          | 0.003      |

Table 2: A summary of the benchmark data sets for the semi-supervised classification experiment.

## 5.2 Active Learning on Graphs

Active learning is a domain that faces the same challenges as semi-supervised learning where labels are scarce and expensive to obtain [47]. In active learning, a subset of unlabelled data points are selected sequentially to be queried according to an acquisition function, with the goal of maximizing the accuracy of the predictive model using significantly fewer labels than would be required if the labelled set were sampled uniformly at random [2]. A motivating example of this problem scenario is in the medical setting where the time of human experts is precious, and the machines must aim to make the best use of the time. Therefore, having a data efficient predictive model that can generalize well with few labels is of critical importance in addition to having a good acquisition function.

In this section, we leverage GGP as the semi-supervised classification model of active learner in graph-based active learning problem [47, 28, 11, 22, 29]. The GGP is paired with the proven $\Sigma$-optimal (SOPT) acquisition function to form an active learner [28]. The SOPT acquisition function is model agnostic in that it only requires the Laplacian matrix of the observed graph and the indices of the labelled nodes in order to identify the next node to query, such that the predictive accuracy of the active learner is maximally increased. The main goal of the active learning experiments is to demonstrate that the GGP can learn better than both the GCN and the Label Propagation model (LP) [48] with very few labelled data points.

Starting with only 1 randomly selected labelled data point (i.e., node), the active learner identifies the next data point to be labelled using the acquisition function. Once the label of the said data point is acquired, the classification model is retrained and its test accuracy is evaluated on the remaining unlabelled data points. In our experiments, the process is repeated until 50 labels are acquired. The experiments are also repeated with 10 different initial labelled data points. In addition to the SOPT acquisition function, we show the results of the same models paired with the random acquisition function (RAND) for comparisons.

The test accuracies with different numbers of labelled data points are presented as learning curves in Figure 2. In addition, we summarize the results numerically using the Area under the Learning Curve (ALC) metric in Table 3. The ALC is normalized to have a maximum value of 1, which corresponds to a hypothetical learner that can achieve $100\%$ test accuracy with only 1 label. The results show that the proposed GGP model is indeed more data efficient than the baselines and can outperform both the GCN and the LP models when labelled data are scarce.

The benchmark data sets for the active learning experiments are the Cora and Citeseer data sets. However, due to technical restriction imposed by the SOPT acquisition function, only the largest connected sub-graph of the data set is used. The restriction reduces the number of nodes in the Cora and Citeseer data sets to $2,485$ and $2,120$ respectively. Both of the data sets were also used as benchmark data sets in [28].

We pre-process the BOW features with TFIDF and apply a linear kernel as the base kernel of the GGP. All parameters are jointly optimized using the ADAM optimizer. The GCN and LP models are trained using the settings recommended in [25] and [28] respectively.

| | Cora | Citeseer |
|---|---|---|
| SOPT-GGP | $0.733 \pm 0.001$ | $0.678 \pm 0.002$ |
| SOPT-GCN | $0.706 \pm 0.001$ | $0.675 \pm 0.002$ |
| SOPT-LP | $0.672 \pm 0.001$ | $0.638 \pm 0.001$ |
| RAND-GGP | $0.575 \pm 0.007$ | $0.557 \pm 0.008$ |
| RAND-GCN | $0.584 \pm 0.011$ | $0.533 \pm 0.008$ |
| RAND-LP | $0.424 \pm 0.020$ | $0.490 \pm 0.011$ |

Table 3: This table shows the Area under the Learning Curve (ALC) scores for the active learning experiments. ALC refers to the area under the learning curves shown in Figure 2 normalized to have a maximum value of 1. The ALCs are computed by averaging over 10 different initial data points. The results show that the GGP is able to generalize better with fewer labels compared to the baselines. 'SOPT' and 'RAND' refer to the acquisition functions used. Please refer to Section 5.2 for discussions of the results.

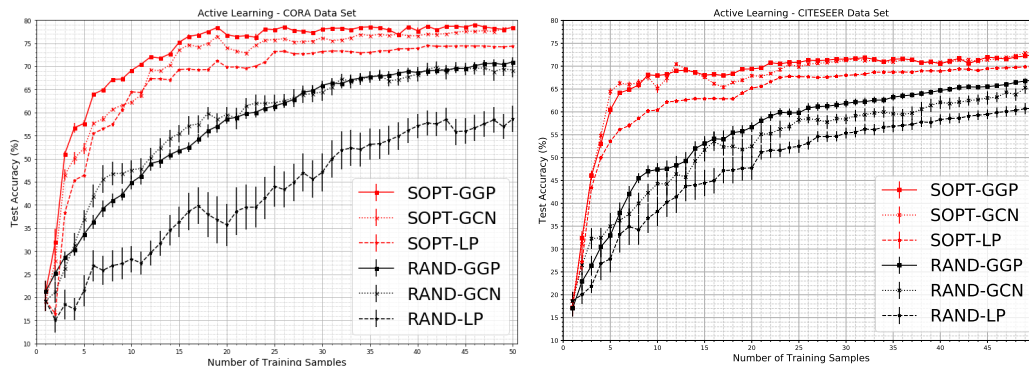

Figure 2: The sub-figures show the test accuracies from the active learning experiments (y-axis) for the Cora (left) and Citeseer (right) data sets with different number of labelled data points (x-axis). The results are averaged over 10 trials with different initial data points. SOPT and RAND refer to the acquisition functions described in Section 5.2. The smaller error bars of 'RAND-GGP' compared to those of 'RAND-GCN' demonstrate the relative robustness of the GGP models under random shuffling of data points in the training data set. The tiny error bars of the 'SOPT-*' results show that the 'SOPT' acquisition function is insensitive to the randomly selected initial labelled data point. Please also refer to Table 3 for numerical summaries of the results.

# 6   Conclusion

We propose a Gaussian process model that is data-efficient for semi-supervised learning problems on graphs. In the experiments, we show that the proposed model is competitive with the state-of-the-art deep learning models, and outperforms when the number of labels is small. The proposed model is simple, effective and can leverage modern scalable variational inference algorithm for GP with minimal modification. In addition, the construction of our model is motivated by distribution regression using the empirical kernel mean embeddings, and can also be viewed under the framework of filtering in the graph spectrum. The spectral view offers a new potential research direction that can be explored in future work.

## Acknowledgements

This work was supported by The Alan Turing Institute under the EPSRC grant EP/N510129/1.

## Footnotes

[1]https://github.com/markvdw/GPflow-inter-domain

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
