[Reviews · NeurIPS 2018]

Reviewer 1



## [Updated after author feedback] Thank you for the feedback. The explanation in L32-L34 is very helpful. I hope you will add it to the main paper. I am happy that figure 2 now has error bars and the note on the small errors on the SOPT-* models is informative. It is good to see that the SOPT-* results are indeed significantly different. I hope, however, that you will also update table 1 in the main paper to show the standard deviations on the 10 random trials. ## Summary The paper presents a Gaussian process-based model for semi-supervised learning on graphs (GGP). The model consists of a novel covariance function, which considers nodes in the 1-hop neighbourhood. The authors further present two distinct views on this new covariance function, namely that of a kernel mean embedding and that of a Bayesian linear model relating to the graph Laplacian. The latter is remarked as being particularly interesting since it can be viewed in terms of a spectral filter, which can be designed to have attractive properties. Finally, the authors cast the model in a variational inference framework with inducing inputs, thus allowing the model to scale to large graphs. ## Quality The paper is of good technical quality. The authors provide discussions and explanations along the way, providing good insights for the reader. The experiments section show a very fair comparison with graph convolutional neural networks by replicating an experiment where these are known to work well. The GGP model is thus compared to nine other methods in a semi-supervised setting. While I like this set-up, I would prefer to have uncertainties on the classification accuracies. There are several competitive methods, all showing promising accuracies, and it is not clear how significant these results are. An additional, interesting experiment shows the method's performance in an active learning setting. This is an interesting experiment, but I would again like to see errorbars on the plots in figure 2 corresponding to the standard deviation over the 10 trials performed. ## Clarity The paper reads well and is easy to follow. The structure makes sense, and figure 1 is very helpful in understanding the method. One section that confused me, however, is lines 156-163. While I understand what you are saying, I am not sure what the implications for your model should be. ## Originality As the authors mention in the (quite extensive) related work section, GPs on graphs have been studied before. The novelty lies in the covariance function, which is simple, yet shown to be effective. By presenting two different views on the developed covariance function, the authors further define interesting directions for future research, which is valuable information for the community. ## Significance I think this paper constitutes a nice addition to the literature on graph Gaussian processes. The proposed covariance function is easy to interpret, yet powerful, and the authors point out interesting research directions. While not groundbreaking novel, it is solid paper that fits NIPS.

Reviewer 2



[post-rebuttal] Thanks for the rebuttal - the change proposed by R1 seem to good to me! The model applies graph Gaussian processes to node classification in a graph. The posed task is to accurately perform classification and active learning in the face of scarce labels. The model compares favourably to various baselines when there are many labels, and outperforms them in the limited data regime. # Overall Score I would be confident implementing this model from the paper, and the experiments seem fair and to present the model's strengths in a balanced way. To my knowledge, the paper attributes prior work well, and makes a clear contribution on top of this work. I believe the paper to be significant in its data efficiency. # Confidence I am familiar with GPs and their variational approximations, and have somewhat out of date familiarity with neural architectures for graph embedding from a project last year, however I cannot claim to have sufficient understanding of the derivations in Section 3 to be able to implement the model without close reference to the paper. Section 3.1 was helpful to my understanding of the paper.

Reviewer 3



This work proposes a Gaussian process on graphs. Given node features, the idea is to first sample a Gaussian process on graph nodes, and then average the outputs of 1-hop neighborhood as the likelihood parameters for each node, to sample the observations. The likelihood parameters can be treated as another Gaussian process, with graph Laplacian incorporated. The paper gives some interesting views of the modeling of the likelihood parameters --- another GP or Bayesian nonlinear model specific feature mappings. The experiments on active learning on graphs and prediction on graphs shows prediction performance comparable to or better than graph convolutional CNN. This is a piece of interesting work that incorporates the graph structure in Gaussian process modeling. The application is interesting as well. The active learning case resembles a little bit Bayesian optimization process and I am not surprised it works well. The paper claims that is a semi-supervised learning work --- however, I am not very convinced. So where does the information of the test data come from? Is it from the graph structure? I hope the authors should explain this more clearly. It seems the experiments do not compare with standard GP, why? The paper lacks a very important baseline --- doing so we can see how the graph structure takes effect. Why do you just use 1-hop average? Is it for simple or for connecting the graph Laplacian? Why don’t use weighted average? Have you considered 2-hop average or even 3-hop average? What will be the benefit or trade-off? I think it will be much better than the paper has discussed or compared with these choices.